ecology, evolution, microbiology

plasmid conjugation, transfer rate, plasmid–host interaction, barriers to genetic exchange

**Author for correspondence:**
Tatiana Dimitriu
e-mail: t.dimitriu@exeter.ac.uk

### PUBLISHING

# Bacteria from natural populations transfer plasmids mostly towards their kin

Tatiana Dimitriu, Lauren Marchant, Angus Buckling and Ben Raymond

Department of Biosciences, University of Exeter, Penryn Campus, Cornwall TR10 9FE, UK

TD, 0000-0002-1604-2622; BR, 0000-0002-3730-0985

Plasmids play a key role in microbial ecology and evolution, yet the determinants of plasmid transfer rates are poorly understood. Particularly, interactions between donor hosts and potential recipients are understudied. Here, we investigate the importance of genetic similarity between naturally co-occurring *Escherichia coli* isolates in plasmid transfer. We uncover extensive variability, spanning over five orders of magnitude, in the ability of isolates to donate and receive two different plasmids, R1 and RP4. Overall, transfer is strongly biased towards clone-mates, but not correlated to genetic distance when donors and recipients are not clone-mates. Transfer is limited by the presence of a functional restriction-modification system in recipients, suggesting sharing of strain-specific defence systems contributes to bias towards kin. Such restriction of transfer to kin sets the stage for longer-term coevolutionary interactions leading to mutualism between plasmids and bacterial hosts in natural communities.

## 1. Introduction

Conjugative plasmids play a central role in horizontal gene transfer, impacting both evolutionary and ecological processes. At large phylogenetic scales, they are the main vector of genetic exchange among bacteria [1], shaping gene flow and long-term adaptation of communities. They also encode a diversity of 'accessory genes' [2] often conferring environment-specific adaptations such as antibiotic and metal resistance and virulence traits. As a consequence, the dynamics of horizontal transfer has crucial consequences for the outcome of competition between lineages, which in turn can both drive the epidemiology of bacterial pathogens and influence ecosystem services. In particular, antibiotic-resistance-conferring plasmid transfer can govern success of strains within patients [3–5] and facilitate pathogen epidemics [6]. An understanding of factors controlling transfer rates is therefore critical.

A striking feature of plasmid transmission by conjugative transfer is its variability. Indeed, estimates of transfer rates lead to fundamentally different conclusions about whether plasmids can naturally persist in the absence of selection on plasmid-carried traits [7–10]. Transfer rates are dependent on both the initiation of conjugation in donor cells and successful establishment in recipient cells [11]. Quantification of transfer rates in the laboratory has mostly focused on a few laboratory strains, which are poor models for natural populations [12], despite the strong effect host genotype and plasmid–host interactions can have on plasmid transfer rates. Transfer rates for plasmid R1 span seven orders of magnitude among natural isolates [13,14]. Plasmids might thus be either lost or spread to fixation depending on host community composition—with rare efficient donors having a particularly strong effect [14]. The detailed pathways of plasmid transfer can also have profound implications. Different plasmid groups are specifically associated with different host lineages, suggesting that barriers to transfer can contribute to global patterns in plasmid host range [15–18]. Moreover, biased transfer of beneficial plasmids towards kin (i.e. between donors and recipients of the same genotype) can favour host bacteria with high investment in transfer, through kin selection

[19]. This in turn could lead to higher transfer for traits including antibiotic resistance under antibiotic selection.

Earlier work showed that bacterial hosts from strain collections of *Escherichia coli* indeed display biased transfer to kin [19]. However, isolates from these collections were distantly related, and unlikely to have coexisted in natural environments. To understand its ecological and evolutionary implications, the diversity in transfer rates among bacteria needs to be characterized within natural populations. Genetic variation in transfer rates, or genetic distance effects, might exist only at a large phylogenetic scale or be the product of environment-specific selective forces, such that strains isolated from the same environment have homogeneous transfer rates or display no bias in transfer towards kin. Here, we investigate the transfer rates of two resistance plasmids, R1 and RP4, with, respectively, narrow and broad host ranges, among a collection of *E. coli* natural isolates, for which population structure and native plasmid content have been characterized previously [16]. *Escherichia coli* from independent populations of grazing cattle were shown to display hierarchical population structure, with high variability in genotypes across cattle populations and individuals, but reduced variability within individuals [16]. Plasmid carriage varied within serotype within individual cattle, suggesting that most opportunity for plasmid transfer was between closely related bacteria. We ask if the striking diversity in transfer previously observed within heterogeneous laboratory strain collections is still present within natural populations, when host bacteria are isolated from the same natural population or are closely related, and we explore genetic factors that could contribute to biased transfer towards kin.

## 2. Methods

### (a) Bacterial strains and plasmids

*Escherichia coli* field isolates (electronic supplementary material, table S1) were selected from the field collection characterized in [16], which surveyed *E. coli* diversity and plasmid content in grazing cattle. Genotypic diversity was previously assessed by sequencing H-antigens classifying *E. coli* into serotypes [20]. Serotypes were highly diverse across individual cattle and cattle populations; by contrast, within-host diversity was reduced, with most cowpats containing only one or two serotypes [16]. To cover *E. coli* diversity present in the collection, we selected 14 strains belonging to seven different serotypes and originating from six different field sites (electronic supplementary material, table S1). Isolates with detected plasmid replicons were excluded. We also used laboratory strains of the *E. coli* K-12 lineage, MG1655 and its derivative MFDpir [21], as standard recipient and donor strains. To test for the effect on transfer rates of restriction-modification (RM) systems, defence systems against foreign DNA [22], we used MG1655 ΔhsdS::Kn generated by P1 transduction from the Keio collection [23].

We characterized transfer rates for two plasmids belonging to the replicon incompatibility groups most abundant in the field collection [16]: IncF, a group of narrow-host-range plasmids only replicating in Enterobacteriaceae, and IncP, a group of broad-host-range plasmids that can transfer and replicate in a wide range of Gram-negative bacteria. R1 plasmid [24] is an IncFII plasmid, in which regulation of transfer is representative of the majority of IncF plasmids [25]. RP4 plasmid [26] is a model IncP-α plasmid. R1 and RP4 were conjugated into unmarked field isolates using the donor strain MFDpir [21], which requires di-aminopimelate (DAP, Sigma-Aldrich) to grow in LB medium.

For all other conjugation assays, spontaneous rifampicin-resistant (Rif$^R$) mutants of MG1655 and the 14 field isolates were generated by plating overnight cultures on LB-agar with rifampicin (Rif, Sigma-Aldrich) at 100 µg ml$^{-1}$, to use as recipients.

### (b) Experimental design and conjugation assays

Conjugation assays were performed by mixing equal volumes of overnight cultures of donors and recipients with 10-fold total dilution into 1 ml LB medium, supplemented with DAP 0.3 mM when MFDpir was used as a donor. Overnight cultures for conjugation experiments did not contain antibiotics. Mixes were incubated at 37°C with 150 r.p.m. shaking. To favour detection of relatively low transfer rates, mating assays were performed for 3 h as a general standard. When stated, assays with a reduced 1 h mating were performed to limit secondary transfer from transconjugants. Donor, recipient and transconjugant densities were then estimated by dilution plating onto selective plates: plasmid-containing bacteria were selected with kanamycin (Kn, 50 µg ml$^{-1}$), except in experiments including MG1655 ΔhsdS::Kn strain and R1 plasmid, for which chloramphenicol (Cm, Sigma-Aldrich, 25 µg ml$^{-1}$) was used instead (see electronic supplementary material, figure S1 for details). Control assays (with donors only or recipients only) never showed any growth on selective transconjugant plates. Each conjugation was performed with at least two biological replicates (separate conjugation assays) per experiment and two independent experiments done on different days.

We first performed conjugation assays from K-12 to the 14 unmarked field isolates, to estimate standard recipient ability and generate plasmid-bearing field isolates. Assays from plasmid-bearing isolates to the Rif$^R$ K-12 recipient then estimated standard donor ability. Next, we used pairs of isolates as donor and recipients (electronic supplementary material, table S1). We first compared 14 kin pairs (where the recipient is the Rif$^R$ derivative of the donor) and 14 non-kin pairs where donor and recipient differ in both serotype and isolation site. For R1 plasmid, we performed additional assays with 14 pairs sharing serotype but isolated from different sites; and 14 pairs from different serotypes isolated from the same site. Finally, to test if RM systems contribute to transfer patterns we compared transfer rates of our field isolates towards the standard K-12 recipient (RM$^+$) and its mutant with no functional type I RM system (RM$^-$), MG1655 ΔhsdS::Kn [27]. RM systems combine a restriction enzyme that cleaves a specific DNA sequence, and a cognate methyl-transferase protecting that same sequence. Foreign DNA originating from cells lacking an RM system present in recipients is not methylated, and thus targeted by restriction. If restriction based on K-12 RM system limits transfer from our natural isolates, we expect transfer rates towards the RM$^-$ mutant to be higher than towards the RM$^+$ recipient.

### (c) Isolate genotyping and phylogenetic distance

We used the phylogenetic markers identified in [28] to study phylogenetic relationships among isolates. The three markers *dinG*, *DPP* and *tonB* were amplified from the 14 field isolates with PCRBio Taq Mix Red (PCR Biosystems) using primers described in [28], and sequenced through Eurofins Genomics. For K-12 strains MG1655 and MFDpir, sequences from GenBank accession NC_000913 were used. Sequence data were pre-processed in GENEIOUS (v. 8.1.6). Amplicons were trimmed at both 3′ and 5′ ends to remove low-quality sequences (i.e. base pairs with an error probability above 5%). High-quality alignments (respectively, 854, 792 and 725 bp long for *dinG*, *DPP* and *tonB*) were concatenated and used to determine multi-locus phylogenetic distance. For isolates D7.8 and oc5.1, *tonB* primers did not yield any product; *DPP* and *dinG* products revealed both isolates were part of *E. marmotae* species. Phylogenetic trees were

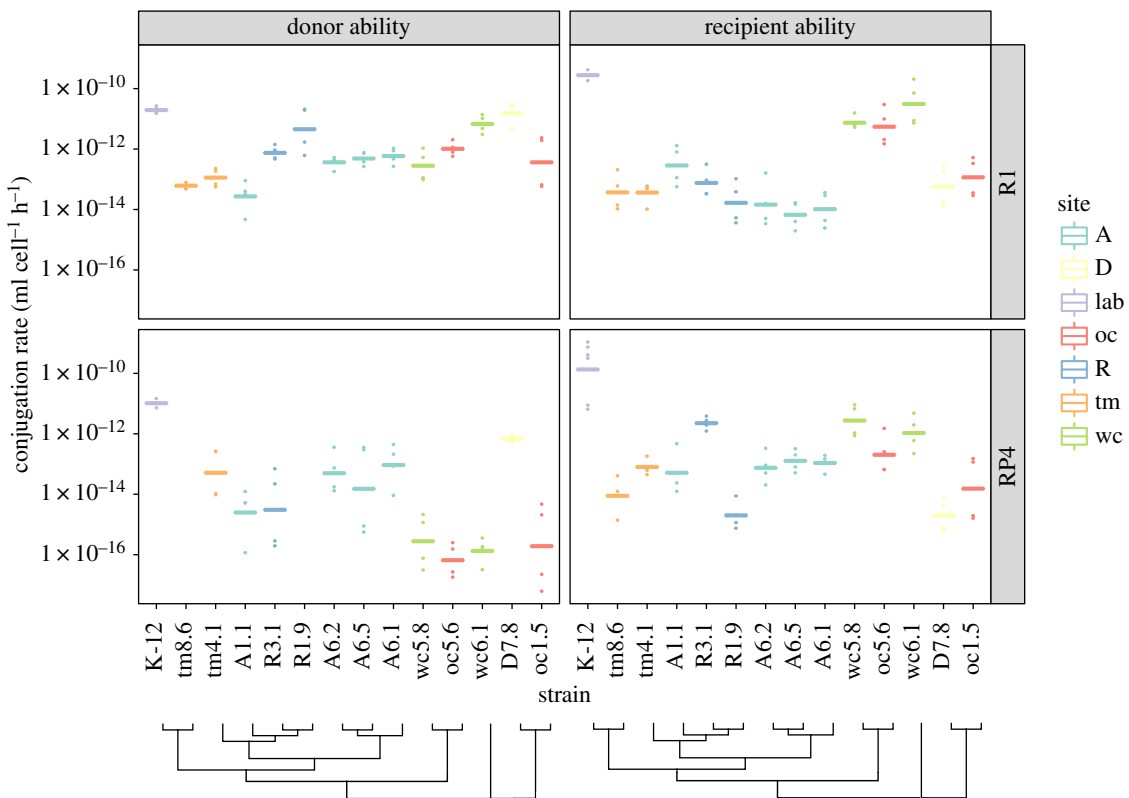

**Figure 1.** Extensive variation in plasmid donor and recipient ability across field isolates of *Escherichia coli*. Conjugation rates were measured in liquid with shaking over 3 h, towards K-12 MG1655 Rif[R] (donor ability) and from K-12 MFDpir (recipient ability). Individual replicates are shown as dots, lines are geometric means. Rates with K-12 as both donor and recipient are shown on the left; field isolates are then ordered by their phylogenetic distance to K-12. Colour indicates site of origin for each isolate. A tree showing phylogenetic relationships is shown under strain names.

made with a weighted neighbour-joining tree building algorithm implemented in GENEIOUS. To obtain phylogenetic distances among all isolates including *E. marmotae*, the tree built with *DPP* and *dinG* products only was used; distances obtained were highly correlated with the ones using all three gene products within *E. coli* isolates (Spearman correlation coefficient $\rho = 0.98$, $p < 2.2 \times 10^{-16}$). Spontaneous Rif[R] mutants were considered identical to the strain they originated from (genetic distance of 0).

### (d) Data analysis

Conjugation rates were measured as $\gamma = T/DRt$ (ml cell$^{-1}$ h$^{-1}$), where $T$, $D$ and $R$, respectively, indicate the density of transconjugants, donors and recipients (cells ml$^{-1}$), and $t$ indicates incubation time (h). When no transconjugants were detected, a threshold conjugation rate was calculated by assuming that one single transconjugant colony was observed. For assays using field isolates as both donors and recipients, variable growth was observed, particularly for recipients (electronic supplementary material, figure S2), probably owing to the spontaneous Rif[R] recipients used in these assays. In order to limit variation in computed conjugation rates owing to low donor or recipient densities, data were excluded when either recipient or donor densities were less than $2 \times 10^7$ cells ml$^{-1}$. As transfer rate values spanned several orders of magnitude, all statistical analysis used log10-transformed data, and averages across replicates were computed as geometric means. For standard donor and recipient ability assays, the effect of field strain identity on transfer rates was tested with one-way ANOVAs. The effect of relationship between donor and recipient within the strain collection was tested with type I ANOVAs as transfer rate~donor identity + recipient identity + relationship. When testing for RM effects, the effect of recipient strain RM status was tested

with a type I ANOVA as transfer rate~donor identity × recipient identity. R v. 3.4.1 was used for all analyses [29].

## 3. Results

### (a) Variation in donor ability and recipient ability across natural isolates

To analyse the amplitude of variation in transfer rates, we first measured transfer rates using K-12 laboratory strains as standard donor or recipient. One of 14 isolates, D2.2 was observed to repeatedly kill K-12 strains in each assay (with less than 1% of K-12 inoculum detected after mating), it was thus excluded from analysis.

For both R1 and RP4 plasmids, transfer rates spanned more than five orders of magnitude overall, from around $10^{-16}$–$10^{-15}$ ml cell$^{-1}$ h$^{-1}$ (detection threshold) to more than $10^{-10}$ ml cell$^{-1}$ h$^{-1}$ (figure 1). For the isolates tm8.6 and R1.9, we were unable to obtain any clones with RP4 across several assays, revealing very low recipient ability; donor ability was not quantified. For both plasmids, donor and recipient ability of K-12 was always as high or higher than any of the field isolates. Excluding K-12, recipient genotype significantly affected conjugation rate from a standard donor for both R1 ($F_{12,39} = 16.05$, $p < 2.10^{-11}$) and RP4 ($F_{12,39} = 13.1$, $p < 10^{-9}$). Similarly, excluding K-12 donor genotype significantly affected conjugation rate towards the standard recipient for both R1 ($F_{12,37} = 11.3$, $p < 10^{-8}$) and RP4 ($F_{10,33} = 8.07$, $p = 2.10^{-6}$). The lowest amplitude of variation was observed for R1 plasmid donor ability, for which all measured rates were above $10^{-14}$ ml cell$^{-1}$ h$^{-1}$. We hypothesized that efficient

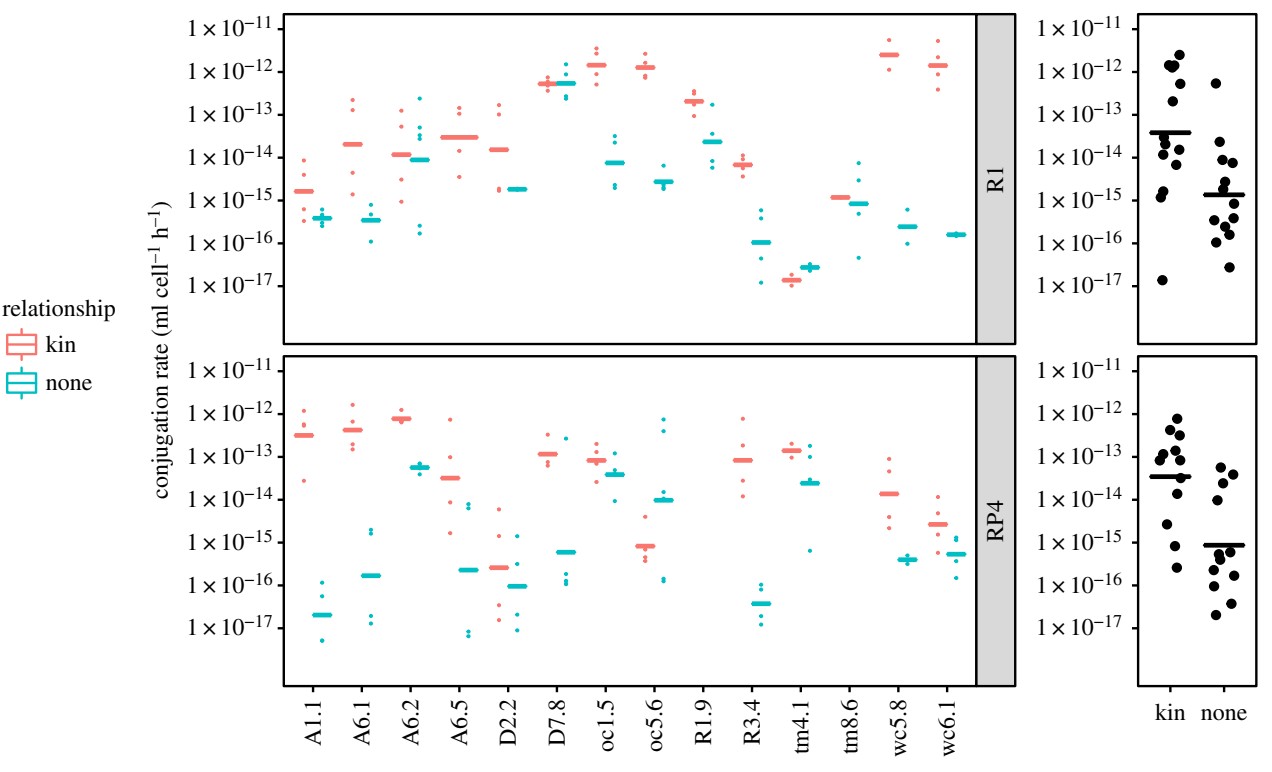

**Figure 2.** Variable transfer rates among field isolates and bias towards kin. Mating assays were performed for 3 h, with donors containing either R1 or RP4 plasmid. Pairs are shown ordered by donor isolate, with the recipient isolate being the same as the donor (kin, red) or another field isolate with distinct serotype and isolation site (non-kin, light blue, see electronic supplementary material, table S1 for recipient identity). Individual replicates are shown as dots, lines are geometric means. Summary graphs at the right show average transfer per couple of strains (dots) and overall geometric means per treatment (lines). Electronic supplementary material, figure S5 presents the same data ordered by recipient isolate.

secondary transfer from K-12 recipients was masking actual variability in transfer from primary donors. Measuring conjugation rates with reduced mating time revealed higher variation among isolates (electronic supplementary material, figure S3) and a stronger effect of the donor ($F_{12,37} = 31.8$, $p < 10^{-15}$) than with longer mating.

No correlation was observed among isolates between average donor and recipient ability (Spearman rank-correlation $\rho = 0.02$, $p = 0.92$); or between average rates between plasmids R1 and RP4 (Spearman rank-correlation $\rho = 0.35$, $p = 0.08$). Moreover, phylogenetic distance from the standard K-12 donor or recipient did not explain average conjugation rates towards or from natural isolates (electronic supplementary material, figure S4; transfer rate~genetic distance $R^2 = 0.003$, $F_{1,48} = 0.14$, $p = 0.71$).

## (b) Comparing transfer among kin and non-kin

We next analysed diversity in transfer rates within our natural populations. To test if transfer is increased towards kin (defined as recipients with strict genetic identity to donors), we performed for each donor conjugation assays to a marked recipient of the same isolate (transfer to kin); and to a randomly chosen isolate both belonging to a different serotype and isolated from a different field site (transfer to non-kin). Rates of transfer were strongly variable across isolates, spanning five orders of magnitude from less than $10^{-6}$ to greater than $10^{-12}$ ml cell$^{-1}$ h$^{-1}$ (figure 2; electronic supplementary material, figure S5). The identity of donor and recipient had a strong effect on conjugation rates (for R1 plasmid, donor effect $F_{13,70} = 20$, $p < 2 \times 10^{-16}$, recipient effect

$F_{9,70} = 10.9$, $p < 3 \times 10^{-10}$; for RP4 plasmid, donor effect $F_{11,74} = 5.93$, $p < 10^{-6}$, recipient effect $F_{7,74} = 11.5$, $p < 10^{-9}$). However, even after accounting for both donor and recipient identity, the relationship between isolates (i.e. being kin or non-kin) was the factor with the largest effect on conjugation rates (for R1, $F_{1,70} = 108$, $p < 10^{-15}$; for RP4, $F_{1,74} = 34.1$, $p < 2.10^{-7}$). On average, a given donor transferred plasmid R1 towards kin 29-fold more efficiently than towards non-kin, and plasmid RP4 40-fold more efficiently. This effect was highly variable across couples, but no isolate was observed to transfer at significantly higher rates towards non-kin. Moreover, higher transfer towards kin could not be explained by an effect of kin on cell densities during competition, as cell densities were not significantly different when donor and recipients were kin or non-kin (one-way ANOVA, donor density~relationship, $F_{1,186} = 0.018$, $p = 0.89$, and recipient density~relationship, $F_{1,186} = 0.32$, $p = 0.57$). There was still high variability among isolates considering only transfer towards clone-mates (transfer rate~strain identity, for R1 $F_{13,35} = 14.1$, $p = 3.10^{-10}$, for RP4 $F_{11,33} = 9.09$, $p < 10^{-6}$), spanning five orders of magnitude. When the same couples of isolates were tested for both plasmids, no correlation in average transfer rates between R1 and RP4 was observed across couples (Pearson correlation coefficient $r_{19} = 0.17$, $p = 0.46$).

## (c) Effect of genetic distance and field site on transfer rates

To understand what leads to the higher conjugation rates observed among clone-mates, we tested if high transfer rates required strict kin identity (genetic distance = 0), or if

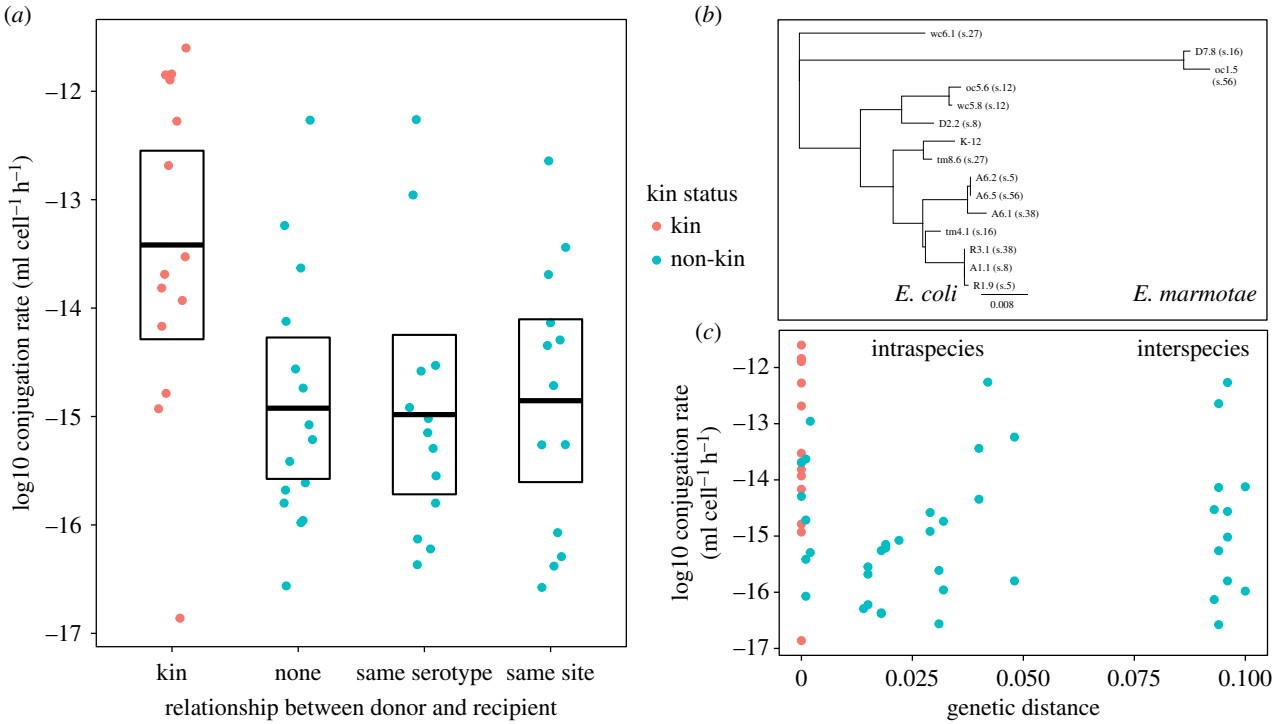

**Figure 3.** Higher transfer rates among field isolates are not correlated with phylogenetic distance but strictly restricted to kin (clone-mates). The average R1 plasmid conjugation rate for a given couple of donor and recipient isolates is shown as a function of the relationship initially defined between donor and recipient (*a*), and of the genetic distance between donor and recipients (*c*). Average rates per couple are shown as dots; boxplots in (*a*) show average rate and 95% confidence interval for each relationship. (*b*) Phylogenetic relationships between 14 field isolates and K-12 laboratory strain, with serotype numbers indicated in brackets.

transfer rates gradually increased with genetic proximity. We focused on the R1 plasmid representative of the IncF plasmid group most abundant in the strain collection, and performed additional conjugation assays choosing couples of isolates that share serotype (initial assessment of their relatedness) or field site of isolation (electronic supplementary material, table S1). We asked if similar variation is observed with shared serotype or within sites, or if isolates from the same field or with same serotype transfer preferentially to each other (figure 3*a*). Overall, the relationship between donors and recipients (i.e. clone-mates, same serotype, same field or no relation) still affected transfer (relationship effect $F_{3,142} = 44.95$, $p < 2 \times 10^{-16}$). However, *post hoc* Tukey tests showed that the only relationship that was significantly different from others was clone-mates, with higher transfer rates ($p < 10^{-7}$ for all comparisons to clone-mates, $p > 0.5$ for all others). Isolates from the same field sites also showed variation in transfer rates (e.g. site R, electronic supplementary material, figure S6), confirming that diversity in transfer rates does occur within the natural populations sampled.

The absence of a serotype effect implies that sharing serotype does not confer high enough relatedness to be equivalent to kin. To understand how genetic distance affects transfer rates more precisely, we derived phylogenetic distance among isolates, which revealed that serotype was a poor indicator of phylogenetic distance (figure 3*b*). Two isolates, D7.8 and oc5.1, were even identified as belonging to *E. marmotae*, despite sharing serotypes with *E. coli* isolates. Overall, there was a small but significant negative effect of phylogenetic distance on conjugation rates (transfer rate~distance, estimate $= -9.03 \pm 2.86$, $r^2 = 0.05$, $p < 0.002$). However, after considering kin/non-kin status, there was no additional

effect of phylogenetic distance (transfer rate~kin status + distance, distance estimate $0.65 \pm 3.14$, $p = 0.836$). Transfer rates were thus not linked to general phylogenetic similarity, but depended only on whether interacting couples were clone-mates or not. Among non-kin, both closely related and inter-species couples had transfer rates spanning from less than $10^{-16}$ to greater than $10^{-13}$ ml cell$^{-1}$ h$^{-1}$ (figure 3*c*), suggesting barriers to transfer can be present even among closely related genotypes, and clone-mates specifically have less barriers to transfer.

## (d) Variation in restriction-modification systems as a mechanism for biased transfer

The restriction of high transfer rates to clone-mates suggests that barriers to transfer are caused by one or few genetic determinants variable at short phylogenetic scales. We tested if variation in RM systems can contribute to the barrier to conjugative transfer in these field isolates, by comparing transfer rates of field isolates towards the standard K-12 recipient (RM$^+$) and an RM$^-$ mutant. We first confirmed that R1 transfer within K-12 is affected by restriction (figure 4, left): as expected, the RM$^+$ strain transfers equally well to both RM$^+$ and RM$^-$ recipients; the RM$^-$ strains transfers at the same rate towards itself, but transfer from the RM$^-$ strain is restricted in RM$^+$ recipients. When measuring transfer from field isolates (figure 4, right), in addition to a strong effect of donor isolate (donor effect $F_{12,106} = 47.2$, $p < 2 \times 10^{-16}$), recipient RM status was also significant ($F_{1,106} = 30.6$, $p < 3 \times 10^{-7}$). On average, the RM$^-$ recipient received R1 plasmid at 3.15-fold higher rates than the RM$^+$ recipient. However, the donor/recipient interaction was significant as well ($F_{12,106} = 2.75$, $p = 0.003$), with only some isolates transferring R1 more efficiently towards

**Figure 4.** The K-12 type I RM system limits transfer from natural isolates. Mating assays were performed for 1 h, from R1 plasmid donors shown on the x-axis towards a K-12 recipient with (RM$^+$, blue) or without (RM$^-$, red) its native RM system. Individual replicates are shown as dots, lines are geometric means. Positive controls with K-12 donors are shown left of the dashed line: deactivating RM in donors decreases conjugation rate when recipients are RM-positive.

the RM$^-$ strain, as expected if some donors also bear an RM system with same specificity as K-12 type I RM. Our results indicate that R1 plasmid is efficiently targeted by restriction, and suggest variation in RM content among field isolates.

## 4. Discussion

We show here that variation in plasmid transfer within *E. coli* isolates from common environments is similar to the variation seen in strain collections [13,14], implying that such variation does not arise from different environment-dependent selective pressures. On the contrary, large differences in transfer rate persist within field site or for closely related isolates. Donor and recipient abilities, as well as transfer rates for R1 and RP4 plasmids, were not correlated, consistent with the different mechanistic basis and regulation of transfer operons in their respective plasmid classes [30]. Interestingly, the broad-host-range plasmid RP4 had similar variation in transfer among hosts, and was no less sensitive to host control than R1, despite suggestions that IncF narrow host range might arise from their more complex regulation by host cells [31]. Moreover, variation in transfer rates among natural isolates might even be higher than estimated here, as we selected isolates with no detected replicons, limiting the effect of modulation of transfer rates by co-resident plasmids [32].

In addition to donor and recipient identity, the main factor controlling transfer rates was the relationship between donors and recipients, with transfer towards kin (clone-mates) being more than 10-fold higher than towards non-kin. We therefore extend the pattern identified previously [19] to a second plasmid, the broad-host-range RP4. The average bias towards kin was even higher for RP4, consistent with the fact that it lacks anti-restriction genes present on R1 [33]. Importantly, we show that bias towards kin occurs among lineages coexisting in the field, indicating that this phenomenon is prevalent in natural populations. Moreover, the effect is restricted to close kin, with no higher transfer towards isolates with relatively closer genetic distance. Thus, discrimination towards kin is

here not a function of average genetic distance among strains [34], but might arise from a combination of few loci [35], likely to be variable even at short genetic distances. Our results are consistent with a study on *Dickeya* strains isolated from the same field site, that despite not being genetically distinguishable using genomic fingerprints, displayed high variation in recipient ability [36].

We identified restriction-modification as a likely mechanism contributing to discrimination in transfer. Restriction was previously shown to limit plasmid conjugation rates with relatively low efficiency [22,37], likely because the first transconjugants escaping restriction are then protected from further restriction when transferring to kin. Similarly, the increase in transfer we observe when inactivating K-12 type I RM is significant but relatively weak in comparison to the strong effect of donor strain. RM systems have tremendously variable target sequence specificity [38], and expression of several systems has a multiplicative effect on restriction efficiency [39], which could amplify the effect we measure with a single system. Our results agree with studies describing the role of RM systems in restricting transfer among lineages [40,41]. As RM systems are very often mobile [42], their transfer among distant strains and loss among closely related strains could explain the large variation in transfer rates independent of genetic distance we observe. The other well-studied defence system of bacteria, CRISPR-Cas, appears less likely to explain our results: targeting of plasmid sequences by recipients could explain some genetic variation in recipient ability [43] but not why transfer is more efficient when plasmids are donated by kin. Some recently discovered mechanisms, however, BREX [44] and DISARM [45], have an epigenetic 'memory' similar to RM systems, which might also contribute to preferential transfer to kin. Finally, other discrimination or structuring processes, not directly targeting plasmid conjugation, would also lead to discrimination in transfer if they affect how much donors encounter kin versus non-kin. This includes non-kin killing by bacteriocins, a form of kin discrimination [46]. Spatial structure, which promotes transfer to kin in the absence of discrimination mechanisms [19], can also bias transfer across a population. Indeed the *E. coli* populations sampled for this study show strong population structure, indicating that opportunities for transfer to plasmid-free isolates occur predominantly within genotypes [16].

The diversity in transfer rates that we uncover has consequences for understanding plasmid maintenance and ecological dynamics. The rates of transfer to kin vary here by five orders of magnitude. These transfer rates within lineages are one of the key determinants of plasmid maintenance [47]. Nine different plasmids were recently shown to be transferred at rates sufficient for persistence, in a classical K-12 strain [10]. Our results suggest that these conclusions should be taken with caution, as natural *E. coli* will probably transfer less than K-12. The scale of variation we observe implies that maintenance of plasmids might depend on subtle details of host genetic composition. Still, a few efficient donors can promote transfer in mixed bacterial populations [14], helping maintaining plasmids in mixed communities [48]. On the other hand, the biased transfer to kin we observe will limit that dynamic, and promote plasmid clustering in distinct lineages. This probably contributes to the variability in plasmid carriage observed among genotypes in the strain collection our strains originate from [16], and in pathogenic

lineages [49]: high transfer rates from efficient donors will be mostly restricted to their own lineage, while strains with low transfer rate might not maintain plasmids efficiently, leading to 'plasmid-shy' genotypes [16]. Moreover, when plasmids confer benefits to their hosts, as with antibiotic exposure for antibiotic resistance plasmids, restricting transfer towards kin will benefit host bacteria and promote indirect selection of efficient donor hosts, through kin selection mechanisms [19]. Transfer towards non-kin, which is efficient for some pairs of isolates in our field collection, might also benefit the hosts when the transferred plasmids bear public-good-encoding genes [50,51], for instance virulence, antibiotic resistance or detoxification genes. More generally, transfer being the highest within kin, together with the observation that plasmids are not at fixation within lineages in the field [16] suggests that most plasmid dynamics might actually occur not between lineages (the events most easily detected) but within lineages, leading to specific coevolution of plasmids with specific host lineages despite recurring dynamics of plasmid transfer and loss.

Data accessibility. The datasets supporting this article are available from the Dryad Digital Repository at https://doi.org/10.5061/dryad.ff045t7 [52].

Authors' contributions. T.D., A.B. and B.R. conceived and designed the research, contributed to revisions. T.D. and L.M. performed the experiments. T.D. performed statistical analysis and wrote the first draft of the manuscript.

Competing interests. We declare we have no competing interests.

Funding. This work was funded by an MRC research council (MR/N013824/1) grant to A.B. and B.R.

Acknowledgements. We thank Chantal Lotton for P1 transduction, and Andrew Matthews for comments.

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
