## [Reviewer comments · Proceedings of the Royal Society B: Biological Sciences]

Review History

RSPB-2019-0805.R0 (Original submission)

Review form: Reviewer 1

Recommendation

Accept with minor revision (please list in comments)

Scientific importance: Is the manuscript an original and important contribution to its field?

Good

General interest: Is the paper of sufficient general interest?

Good

Quality of the paper: Is the overall quality of the paper suitable?

Excellent

Is the length of the paper justified?

Yes

Should the paper be seen by a specialist statistical reviewer?

No

Do you have any concerns about statistical analyses in this paper? If so, please specify them explicitly in your report.

No

It is a condition of publication that authors make their supporting data, code and materials available - either as supplementary material or hosted in an external repository. Please rate, if applicable, the supporting data on the following criteria.

Is it accessible?

Yes

Is it clear?

Yes

Is it adequate?

Yes

Do you have any ethical concerns with this paper?

No

Comments to the Author

I have enjoyed reading the manuscript "Bacteria from natural populations transfer plasmids mostly towards their kin" by Dimitriu et al. This manuscript describes a series of experiments designed to evaluate whether there are biases in the rates of conjugal transfer of plasmids between related bacteria isolated from a common environment. The reported data supports the conclusion that plasmids are more likely to be transferred between clone-mates than between distinct isolates; however, there was not a relationship between the frequency of transfer and the genetic distance of the strains. The authors also provide some data supporting a role of restriction-modification systems in determining the frequency of transfer, consistent with past studies. The reported data has implications for our understanding of plasmid transfer and maintenance dynamics in natural populations, which is important for understanding the spread of plasmids and the genes they host (such as antimicrobial genes).

Overall, I found the manuscript to be well-written and an engaging read. I thought the experiments were well-designed, and that the conclusions were supported by the results. I only have two main comments, neither of which are expected to influence the conclusions or interpretation of the data.

Line 249-260 (and the corresponding section of the discussion): I think that the authors have done an appropriate job of choosing their words to ensure that their conclusion is consistent with the data. However, I have two questions related to this data for the authors to consider. Is there a correlation between the frequency of plasmid transfer to the RM+ and RM- recipients? I.e., do donors with high/low plasmid transfer frequency to the RM+ strain also tend to have high/low transfer frequency to the RM- strain? If so, would this suggest that the hsdS restriction system plays a relatively minor role in determining the frequency of transfer in the tested pairs? I am also curious if there is a relationship between the fold change in conjugation frequency between the RM+ and RM- strains with the frequency of transfer to the RM+ strain. In other words, do donors with a high plasmid transfer frequency to the RM+ recipient tend to have a bigger or smaller changes in frequency (compared to donors with a low transfer frequency) when switching to the RM- recipient? I think this would also shed some light on how important of a role hsdS played in

determining plasmid transfer frequencies, in the given tests. I accept that these questions are perhaps tangent to the main point of this section, and thus the authors may prefer not to address these points within the manuscript.

I am unsure if box plots are the best way to present the data in Figures 1, 2, and 4. With only 4 data points per condition, I wonder how meaningful the box plots, and the quantile calculations, are. Have the authors considered plotting just the data points (as they already do) and possibly a mean line (as in the right side of Figure 2), colouring the dots and the lines as necessary?

In Figures 1, 2, and 4, a few of the dots are large. Could the authors please clarify the meaning of the large dots in the legends (outliers?).

Line 289: Others have also shown that mutation of restriction modification systems increases the efficiency of conjugal transfer in other organisms, such as *Sinorhizobium meliloti* (doi: 10.1016/j.jbiotec.2016.06.033).

Line 64: Should there be a comma after "distance effects"?

Figure S5: The figure legend mentions that the colour indicates kin versus non-kin, whereas the legend on the right side of the figure indicates kin versus site. Should "site" be "non-kin"?

Line 54: The comma after "host range" can be removed, and the citations can be combined [i.e., (15-18) instead of (15-17) (18)].

Review form: Reviewer 2

Recommendation

Accept as is

Scientific importance: Is the manuscript an original and important contribution to its field?

Good

General interest: Is the paper of sufficient general interest?

Good

Quality of the paper: Is the overall quality of the paper suitable?

Good

Is the length of the paper justified?

No

Should the paper be seen by a specialist statistical reviewer?

No

Do you have any concerns about statistical analyses in this paper? If so, please specify them explicitly in your report.

No

It is a condition of publication that authors make their supporting data, code and materials available - either as supplementary material or hosted in an external repository. Please rate, if applicable, the supporting data on the following criteria.

Is it accessible?

Yes

Is it clear?

No

Is it adequate?

Yes

Do you have any ethical concerns with this paper?

No

Comments to the Author

This study looks at variability in the rate of plasmid transfer within a natural community of E coli and related species. While host range has been looked at a great deal, this is often not considered in terms of conjugation rate. Given that gene transfer is occurring on ecological time scales, this is very relevant to our understanding of how communities evolve.

The study is well presented and from what I can tell suitably analysed. I believe it will make a useful addition to the field and have no major comments!

Decision letter (RSPB-2019-0805.R0)

02-May-2019

Dear Dr Dimitriu:

I am writing to inform you that your manuscript RSPB-2019-0805 entitled "Bacteria from natural populations transfer plasmids mostly towards their kin." has, in its current form, been rejected for publication in Proceedings B.

This action has been taken after considering the advice of referees and the Associate Editor. We would be happy to consider a resubmission, provided the comments of the referees are fully addressed. However please note that this is not a provisional acceptance.

Sincerely,

Proceedings B

Associate Editor

Board Member: 1

Comments to Author:

This paper tackles an understudied but very important fundamental question in bacterial evolution and genetics: what determines horizontal gene transfer (HGT), and how does genetic variation within a species affect HGT success. This question has big implications for bacterial evolution very generally, with applied importance in human health, agriculture, and beyond. Combining phylogenetic analysis of natural isolates with conjugation assays to test for variation in plasmid transfer rates is creative. Two reviewers, both experts in the field, were enthusiastic about this manuscript, though reviewer 1 has some useful suggestions to consider. I also reviewed the paper in detail, and have additional comments that I believe will help the paper better connect with the broad readership of Proceedings B.

My major comments relate to 1) framing these detailed experiments in the ecology of the strains, and 2) readability of the paper, so non-specialists can more easily tell what was done and why:

- 1) The premise of this paper is that it expands on previous work to include a more ecologically relevant collection of strains, but the strain collections and population biology of E coli are not introduced in any depth. The sampling scheme (spatial scale, relevance to spatial genetic structure of E coli, etc) are not provided in the intro or methods.
- 2) There are a number of readability issues that could be improved pretty easily. For example, the experimental design should be more linearly written out, so the reader can tell what was actually done (e.g., we performed two replicates of each combination of K12 as donor to all 14 focal strains, then vice versa...). Ideally the methods can be structured to mirror the results so the methods that lead to each set of results can be easily discriminated. Right now it's not clear how many separate experiments were actually performed. Right now the strains are described and the methods of conjugation are described, but the paper is missing an organized explanation of the experimental design.

Line-by-line:

36: consequences for the outcome

77: this section can include more ecological context for the strains. What is the local scale for E coli, what is already known about E coli population structure?

85: standard - what does this mean here?

86-87: because... (provide rationale that explains how these strains allow this test, for the uninitiated? Right now the methods are difficult to approach for a non-specialist already familiar with these E coli strains and the RM systems)

168: K12 vs. MG1655 – these are the same? Not consistent - sometimes one name is used, sometimes the other. Again confusing for a reader who doesn't know the strains.

187: it is somewhat unusual to have a statistical model in the results rather than in the analysis section of the methods – this is tied to the larger issue of the experimental design(s) (and corresponding statistical models) not being clear to the reader during the methods section

191: largest effect?

202: couples?

206: Is this the result of a whole new experiment, or just a new way of parsing the data from the set of conjugations in Figure 2? Again, I believe the methods section can much more clearly explain how many experiments were performed and the design of each.

224-226: How are we supposed to see this lack of correlation in 3B? Serotypes are not indicated

239-248: This paragraph is in the results, but currently conveys background information that is actually necessary for non-specialists to understand the experimental designs (mutants strains, etc) in the methods (which occurs before this section). Setting up these questions before the methods can, I believe, improve readability

286: using a handful of phylogenetically-informative loci/MLST?

295: What is variable about these "systems" – the sequence targeted and the enzymes? Can you clarify?

306: that => which

317: community composition, or genetic composition?

333: leading "to" specific

490: "as a function"

Reviewer(s)' Comments to Author:

Referee: 1

Comments to the Author(s)

I have enjoyed reading the manuscript "Bacteria from natural populations transfer plasmids mostly towards their kin" by Dimitriu et al. This manuscript describes a series of experiments designed to evaluate whether there are biases in the rates of conjugal transfer of plasmids between related bacteria isolated from a common environment. The reported data supports the conclusion that plasmids are more likely to be transferred between clone-mates than between distinct isolates; however, there was not a relationship between the frequency of transfer and the genetic distance of the strains. The authors also provide some data supporting a role of restriction-modification systems in determining the frequency of transfer, consistent with past studies. The reported data has implications for our understanding of plasmid transfer and maintenance dynamics in natural populations, which is important for understanding the spread of plasmids and the genes they host (such as antimicrobial genes).

Overall, I found the manuscript to be well-written and an engaging read. I thought the experiments were well-designed, and that the conclusions were supported by the results. I only have two main comments, neither of which are expected to influence the conclusions or interpretation of the data.

Line 249-260 (and the corresponding section of the discussion): I think that the authors have done an appropriate job of choosing their words to ensure that their conclusion is consistent with the data. However, I have two questions related to this data for the authors to consider. Is there a correlation between the frequency of plasmid transfer to the RM+ and RM- recipients? I.e., do

donors with high/low plasmid transfer frequency to the RM+ strain also tend to have high/low transfer frequency to the RM- strain? If so, would this suggest that the hsdS restriction system plays a relatively minor role in determining the frequency of transfer in the tested pairs? I am also curious if there is a relationship between the fold change in conjugation frequency between the RM+ and RM- strains with the frequency of transfer to the RM+ strain. In other words, do donors with a high plasmid transfer frequency to the RM+ recipient tend to have a bigger or smaller changes in frequency (compared to donors with a low transfer frequency) when switching to the RM- recipient? I think this would also shed some light on how important of a role hsdS played in determining plasmid transfer frequencies, in the given tests. I accept that these questions are perhaps tangent to the main point of this section, and thus the authors may prefer not to address these points within the manuscript.

I am unsure if box plots are the best way to present the data in Figures 1, 2, and 4. With only 4 data points per condition, I wonder how meaningful the box plots, and the quantile calculations, are. Have the authors considered plotting just the data points (as they already do) and possibly a mean line (as in the right side of Figure 2), colouring the dots and the lines as necessary?

In Figures 1, 2, and 4, a few of the dots are large. Could the authors please clarify the meaning of the large dots in the legends (outliers?).

Line 289: Others have also shown that mutation of restriction modification systems increases the efficiency of conjugal transfer in other organisms, such as *Sinorhizobium meliloti* (doi: 10.1016/j.jbiotec.2016.06.033).

Line 64: Should there be a comma after "distance effects"?

Figure S5: The figure legend mentions that the colour indicates kin versus non-kin, whereas the legend on the right side of the figure indicates kin versus site. Should "site" be "non-kin"?

Line 54: The comma after "host range" can be removed, and the citations can be combined [i.e., (15-18) instead of (15-17) (18)].

Referee: 2

Comments to the Author(s)

This study looks at variability in the rate of plasmid transfer within a natural community of *E. coli* and related species. While host range has been looked at a great deal, this is often not considered in terms of conjugation rate. Given that gene transfer is occurring on ecological time scales, this is very relevant to our understanding of how communities evolve.

The study is well presented and from what I can tell suitably analysed. I believe it will make a useful addition to the field and have no major comments!

Author's Response to Decision Letter for (RSPB-2019-0805.R0)

See Appendix A.

RSPB-2019-1110.R0

Review form: Reviewer 1

Recommendation

Accept as is

Scientific importance: Is the manuscript an original and important contribution to its field?

Excellent

General interest: Is the paper of sufficient general interest?

Excellent

Quality of the paper: Is the overall quality of the paper suitable?

Excellent

Is the length of the paper justified?

Yes

Should the paper be seen by a specialist statistical reviewer?

No

Do you have any concerns about statistical analyses in this paper? If so, please specify them explicitly in your report.

No

It is a condition of publication that authors make their supporting data, code and materials available - either as supplementary material or hosted in an external repository. Please rate, if applicable, the supporting data on the following criteria.

Is it accessible?

Yes

Is it clear?

Yes

Is it adequate?

Yes

Do you have any ethical concerns with this paper?

No

Comments to the Author

This is my second time reviewing the manuscript "Bacteria from natural populations transfer plasmids mostly towards their kin" by Dimitriu et al. In my opinion, the authors have appropriately addressed all of the comments raised during the previous round of review, and I have no further comments.

Decision letter (RSPB-2019-1110.R0)

28-May-2019

Dear Dr Dimitriu

I am pleased to inform you that your Review manuscript RSPB-2019-1110 entitled "Bacteria from natural populations transfer plasmids mostly towards their kin." has been accepted for publication in Proceedings B.

The referee(s) do not recommend any further changes. Therefore, please proof-read your manuscript carefully and upload your final files for publication. Because the schedule for publication is very tight, it is a condition of publication that you submit the revised version of your manuscript within 7 days. If you do not think you will be able to meet this date please let me know immediately.

To upload your manuscript, log into <http://mc.manuscriptcentral.com/prsb> and enter your Author Centre, where you will find your manuscript title listed under "Manuscripts with Decisions." Under "Actions," click on "Create a Revision." Your manuscript number has been appended to denote a revision.

You will be unable to make your revisions on the originally submitted version of the manuscript. Instead, upload a new version through your Author Centre.

1) A text file of the manuscript (doc, txt, rtf or tex), including the references, tables (including captions) and figure captions. Please remove any tracked changes from the text before submission. PDF files are not an accepted format for the "Main Document".

2) A separate electronic file of each figure (tiff, EPS or print-quality PDF preferred). The format should be produced directly from original creation package, or original software format. Please note that PowerPoint files are not accepted.

3) Electronic supplementary material: this should be contained in a separate file from the main text and the file name should contain the author's name and journal name, e.g. `authorname_procb_ESM_figures.pdf`

All supplementary materials accompanying an accepted article will be treated as in their final form. They will be published alongside the paper on the journal website and posted on the online figshare repository. Files on figshare will be made available approximately one week before the accompanying article so that the supplementary material can be attributed a unique DOI. Please see: <https://royalsociety.org/journals/authors/author-guidelines/>

4) Data-Sharing and data citation

It is a condition of publication that data supporting your paper are made available. Data should be made available either in the electronic supplementary material or through an appropriate repository. Details of how to access data should be included in your paper. Please see <https://royalsociety.org/journals/ethics-policies/data-sharing-mining/> for more details.

<http://datadryad.org/submit?journalID=RSPB&manu=RSPB-2019-1110> which will take you to your unique entry in the Dryad repository.

Once again, thank you for submitting your manuscript to Proceedings B and I look forward to receiving your final version. If you have any questions at all, please do not hesitate to get in touch.

Sincerely,

Proceedings B
mailto:proceedingsb@royalsociety.org

Associate Editor
Board Member
Comments to Author:
(There are no comments.)

Reviewer(s)' Comments to Author:

Referee: 1

Comments to the Author(s).

This is my second time reviewing the manuscript "Bacteria from natural populations transfer plasmids mostly towards their kin" by Dimitriu et al. In my opinion, the authors have appropriately addressed all of the comments raised during the previous round of review, and I have no further comments.

Sincerely,

Proceedings B
mailto:proceedingsb@royalsociety.org

Decision letter (RSPB-2019-1110.R1)

31-May-2019

Dear Dr Dimitriu

I am pleased to inform you that your manuscript entitled "Bacteria from natural populations transfer plasmids mostly towards their kin." has been accepted for publication in Proceedings B.

You can expect to receive a proof of your article from our Production office in due course, please check your spam filter if you do not receive it. PLEASE NOTE: you will be given the exact page

length of your paper which may be different from the estimation from Editorial and you may be asked to reduce your paper if it goes over the 10 page limit.

Open Access

Paper charges

Sincerely,

Appendix A

Associate Editor

Board Member: 1

Comments to Author:

This paper tackles an understudied but very important fundamental question in bacterial evolution and genetics: what determines horizontal gene transfer (HGT), and how does genetic variation within a species affect HGT success. This question has big implications for bacterial evolution very generally, with applied importance in human health, agriculture, and beyond. Combining phylogenetic analysis of natural isolates with conjugation assays to test for variation in plasmid transfer rates is creative. Two reviewers, both experts in the field, were enthusiastic about this manuscript, though reviewer 1 has some useful suggestions to consider. I also reviewed the paper in detail, and have additional comments that I believe will help the paper better connect with the broad readership of Proceedings B.

We thank the editor and reviewers for their very helpful comments and have answered each point as detailed below.

My major comments relate to 1) framing these detailed experiments in the ecology of the strains, and 2) readability of the paper, so non-specialists can more easily tell what was done and why:

1) The premise of this paper is that it expands on previous work to include a more ecologically relevant collection of strains, but the strain collections and population biology of E coli are not introduced in any depth. The sampling scheme (spatial scale, relevance to spatial genetic structure of E coli, etc) are not provided in the intro or methods.

We have added relevant information on the strain collection, its diversity and sampling scheme to the methods, and briefly mentioned it in the introduction.

2) There are a number of readability issues that could be improved pretty easily. For example, the experimental design should be more linearly written out, so the reader can tell what was actually done (e.g., we performed two replicates of each combination of K12 as donor to all 14 focal strains, then vice versa...). Ideally the methods can be structured to mirror the results so the methods that lead to each set of results can be easily discriminated. Right now it's not clear how many separate experiments were actually performed. Right now the strains are described and the methods of conjugation are described, but the paper is missing an organized explanation of the experimental design. We have added an overall explanation of the experimental design in the methods, following the results linearly. We also added a supplementary figure (Figure S1) to make clear which combinations of strains were used for each experiment.

Line-by-line:

36: consequences for the outcome done

77: this section can include more ecological context for the strains. What is the local scale for E coli, what is already known about E coli population structure? We have added this information.

85: standard – what does this mean here? We removed this term from the strain description, here and line 98, as we meant it as part of the experimental design (meaning a fixed donor /recipient).

86-87: because... (provide rationale that explains how these strains allow this test, for the uninitiated? Right now the methods are difficult to approach for a non-specialist already familiar with these E coli strains and the RM systems) We added the rationale and moved some of our results explanations to the methods, as suggested below.

168: K12 vs. MG1655 – these are the same? Not consistent - sometimes one name is used, sometimes the other. Again confusing for a reader who doesn't know the strains. We have changed MG1655 / MFDpir to K-12, the original laboratory lineage name, everywhere where

lineage information is relevant. We've kept MG1655 / MFDpir only in the methods, where needed to describe the precise strain used.

187: it is somewhat unusual to have a statistical model in the results rather than in the analysis section of the methods – this is tied to the larger issue of the experimental design(s) (and corresponding statistical models) not being clear to the reader during the methods section We have moved this statistical model and other equivalent ones to the data analysis section of the methods, following linearly the section on experimental design.

191: largest effect? done

202: couples? done

206: Is this the result of a whole new experiment, or just a new way of parsing the data from the set of conjugations in Figure 2? It uses data from a new experiment to consider serotype and field couples, which are compared to the R1 data (kin and non-kin) also in Figure 2. We now say that explicitly in the results. Again, I believe the methods section can much more clearly explain how many experiments were performed and the design of each. done

224-226: How are we supposed to see this lack of correlation in 3B? Serotypes are not indicated. We have added serotype numbers to figure 3B.

239-248: This paragraph is in the results, but currently conveys background information that is actually necessary for non-specialists to understand the experimental designs (mutants strains, etc) in the methods (which occurs before this section). Setting up these questions before the methods can, I believe, improve readability We moved this background information to the methods.

286: using a handful of phylogenetically-informative loci/MLST? done

295: What is variable about these “systems” – the sequence targeted and the enzymes? Can you clarify? We clarified that variation is indeed in target sequence specificity, and also separated the two arguments about variability and horizontal transfer for more clarity.

306: that => which done

317: community composition, or genetic composition? done

333: leading “to” specific done

490: “as a function” done

Reviewer(s)' Comments to Author:

Referee: 1

Comments to the Author(s)

I have enjoyed reading the manuscript "Bacteria from natural populations transfer plasmids mostly towards their kin" by Dimitriu et al. This manuscript describes a series of experiments designed to evaluate whether there are biases in the rates of conjugal transfer of plasmids between related bacteria isolated from a common environment. The reported data supports the conclusion that plasmids are more likely to be transferred between clone-mates than between distinct isolates; however, there was not a relationship between the frequency of transfer and the genetic distance of the strains. The authors also provide some data supporting a role of restriction-modification systems in determining the frequency of transfer, consistent with past studies. The reported data has implications for our understanding of plasmid transfer and maintenance dynamics in natural populations, which is important for understanding the spread of plasmids and the genes they host (such as antimicrobial genes).

Overall, I found the manuscript to be well-written and an engaging read. I thought the experiments were well-designed, and that the conclusions were supported by the results. I only have two main comments, neither of which are expected to influence the conclusions or interpretation of the data.

Line 249-260 (and the corresponding section of the discussion): I think that the authors have

done an appropriate job of choosing their words to ensure that their conclusion is consistent with the data. However, I have two questions related to this data for the authors to consider. Is there a correlation between the frequency of plasmid transfer to the RM+ and RM- recipients? I.e., do donors with high/low plasmid transfer frequency to the RM+ strain also tend to have high/low transfer frequency to the RM- strain? If so, would this suggest that the hsdS restriction system plays a relatively minor role in determining the frequency of transfer in the tested pairs? I am also curious if there is a relationship between the fold change in conjugation frequency between the RM+ and RM- strains with the frequency of transfer to the RM+ strain. In other words, do donors with a high plasmid transfer frequency to the RM+ recipient tend to have a bigger or smaller changes in frequency (compared to donors with a low transfer frequency) when switching to the RM- recipient? I think this would also shed some light on how important of a role hsdS played in determining plasmid transfer frequencies, in the given tests. I accept that these questions are perhaps tangent to the main point of this section, and thus the authors may prefer not to address these points within the manuscript.

We thank the reviewer for these comments. We would indeed like to estimate and investigate the variability in hsdS effect depending on donor isolates, however we did not include these analyses as we were sceptical about these data having enough replication to confidently conclude about individual strain effects. We limited ourselves to analysing overall effects.

There is indeed a positive correlation between average transfer from a donor strain to the RM+ and RM- strains (Pearson's correlation coefficient = 0.92, $p=10^{-5}$), but this can be obtained only working with averages, as replicates for RM+ and RM- recipients cannot be meaningfully paired. Instead, we pointed to the same general conclusion in the statistical analysis described in the results, as the ANOVA does show a strong effect of donor strain identity on transfer rates, suggesting similarly that the main effect explaining variation in transfer is the presence of high transfer donors and low transfer donors. To emphasize this point, we have now added a comment on hsdS effect being relatively weak in comparison to donor strain effect in the discussion.

Following the reviewer's suggestion, we did find a trend towards negative correlation between fold-change in conjugation rate between RM+ and RM- strain and transfer rate towards the RM+ strain, however this effect is not significant (Pearson's correlation coefficient $r = 0.5$, $p=0.08$). We chose to not add these results to our manuscript, as the differences calculated here between average rates per strain obtained with limited replications are likely not very accurate.

I am unsure if box plots are the best way to present the data in Figures 1, 2, and 4. With only 4 data points per condition, I wonder how meaningful the box plots, and the quantile calculations, are. Have the authors considered plotting just the data points (as they already do) and possibly a mean line (as in the right side of Figure 2), colouring the dots and the lines as necessary? We thank the reviewer for the suggestion and have made these changes for those figures, as well as the corresponding supplementary figures.

In Figures 1, 2, and 4, a few of the dots are large. Could the authors please clarify the meaning of the large dots in the legends (outliers?). They were indeed outliers in respect to the boxplots, and do not appear anymore.

Line 289: Others have also shown that mutation of restriction modification systems increases the efficiency of conjugal transfer in other organisms, such as *Sinorhizobium meliloti* (doi: 10.1016/j.jbiotec.2016.06.033). Thanks, we have added that reference.

Line 64: Should there be a comma after "distance effects"? yes, done.

Figure S5: The figure legend mentions that the colour indicates kin versus non-kin, whereas the legend on the right side of the figure indicates kin versus site. Should "site" be "non-kin"?

Indeed, thanks for spotting this. They are part of the same site, but the relevant information here is non-kin.

Line 54: The comma after "host range" can be removed, and the citations can be combined [i.e., (15-18) instead of (15-17) (18)]. Done.

Referee: 2

Comments to the Author(s)

This study looks at variability in the rate of plasmid transfer within a natural community of *E. coli* and related species. While host range has been looked at a great deal, this is often not considered in terms of conjugation rate. Given that gene transfer is occurring on ecological time scales, this is very relevant to our understanding of how communities evolve.

The study is well presented and from what I can tell suitably analysed. I believe it will make a useful addition to the field and have no major comments!

We thank the reviewer for their nice comments!